REGISTERED REPORT PROTOCOL

# Speech-in-noise, psychosocial, and heart rate variability outcomes of group singing or audiobook club interventions for older adults with unaddressed hearing loss: A SingWell Project multisite, randomized controlled trial, registered report protocol

Chi Yhun Lo[1]*, Benjamin Rich Zendel[2], Deniz Baskent[3], Christian Boyle[4], Emily Coffey[5], Nathan Gagne[5], Assal Habibi[6], Ellie Harding[7], Merel Keijzer[7], Gunter Kreutz[8], Bert Maat[9], Eva Schurig[8], Mridula Sharma[4], Carmen Dang[1], Sean Gilmore[1], Helen Henshaw[10], Colette M. McKay[11], Arla Good[1], Frank A. Russo[1]

**1** Department of Psychology, Toronto Metropolitan University, Toronto, ON, Canada, **2** Faculty of Medicine, Memorial University of Newfoundland, St John's, NL, Canada, **3** Faculty of Medicine, University of Groningen, Groningen, GR, Netherlands, **4** College of Nursing and Health Sciences, Flinders University, Adelaide, SA, Australia, **5** Department of Psychology, Concordia University, Montreal, QC, Canada, **6** Brain and Creativity Institute, University of Southern California, Los Angeles, CA, United States of America, **7** Faculty of Arts, University of Groningen, Groningen, GR, Netherlands, **8** Institute of Music, Carl von Ossietzky University of Oldenburg, Oldenburg, NI, Germany, **9** Department of Otorhinolaryngology, University of Groningen, Groningen, GR, Netherlands, **10** NIHR Nottingham Biomedical Research Centre, Hearing Sciences, School of Medicine, Mental Health and Clinical Neurosciences, University of Nottingham, Nottingham, United Kingdom, **11** Bionics Institute, Melbourne, VIC, Australia

* chi.lo@torontomu.ca

## Abstract

### Background

Unaddressed age-related hearing loss is highly prevalent among older adults, typified by negative consequences for speech-in-noise perception and psychosocial wellbeing. There is promising evidence that group singing may enhance speech-in-noise perception and psychosocial wellbeing. However, there is a lack of robust evidence, primarily due to the literature being based on small sample sizes, single site studies, and a lack of randomized controlled trials. Hence, to address these concerns, this SingWell Project study utilizes an appropriately powered sample size, multisite, randomized controlled trial approach, with a robust preplanned statistical analysis.

### Objective

To explore if group singing may improve speech-in-noise perception and psychosocial wellbeing for older adults with unaddressed hearing loss.

### Methods

We designed an international, multisite, randomized controlled trial to explore the benefits of group singing for adults aged 60 years and older with unaddressed hearing loss (registered

**Data Availability Statement:** All relevant data from this study will be made available upon study completion.

**Funding:** Funding for this study is provided through a Social Sciences and Humanities Research Council of Canada (SSHRC) Partnership Grant awarded to F. Russo (Reference Number: 895-2021-1018) The funders had no role in study design, data collection and analysis, decision to publish, or preparation of the manuscript.

**Competing interests:** The authors have declared that no competing interests exist.

at clinicaltrials.gov, ID: NCT06580847). After undergoing an eligibility screening process and completing an information and consent form, we intend to recruit 210 participants that will be randomly assigned to either group singing or an audiobook club (control group) intervention for a training period of 12-weeks. The study has multiple timepoints for testing, that are broadly categorized as macro (i.e., pre- and post-measures across the 12-weeks), or micro timepoints (i.e., pre- and post-measures across a weekly training session). Macro measures include behavioural measures of speech and music perception, and psychosocial questionnaires. Micro measures include psychosocial questionnaires and heart-rate variability.

## Hypotheses

We hypothesize that group singing may be effective at improving speech perception and psychosocial outcomes for adults aged 60 years and older with unaddressed hearing loss—more so than participants in the control group.

## Introduction

### Age-related hearing loss and its consequences

Hearing loss is common and currently affects 1.6 billion people worldwide [1]. The consequences of hearing loss have a significant impact on individual quality of life and place a considerable burden on global health and economic systems [2], with a conservative cost-analysis estimated at $981 billion [3]. This is exacerbated due to a rapidly ageing population, with hearing loss projected to extend globally to 2.1 billion people by 2050 [4]. Age-related hearing loss (ARHL, or presbycusis) is a progressive and bilateral sensorineural hearing loss that impacts both central and peripheral auditory functions, and disproportionately affects higher frequencies [5, 6]. ARHL is highly prevalent amongst older adults, with the estimated incidence doubling for every ten-year increase in age [7]. For hearing losses greater than 25 dB (equivalent to a mild hearing loss or worse), the estimated incidence varies from 42.3% to 63.1% for adults aged 70 years and older, depending on test context and population factors [7, 8].

ARHL is associated with communication challenges, which is typically exemplified by difficulties perceiving speech in the presence of noise [9–11]. This is significant given the congregation of humans at social activities and experiences such as dining-out at eateries are typified by noisy sound propagation. Real world noise levels indicate that older adults with mild to moderate hearing loss spend approximately 7.5% of their time in listening situations where noise meets or exceeds speech levels [12]. Importantly, while speech-in-noise (SIN) tasks used in clinical and research contexts are typically receptive in nature (i.e., focussed on individual ability to perceive SIN); the real-world outcomes are far broader, with significant social and emotional impacts that extend to communication partners and beyond [13, 14]. The ability to communicate effectively is fundamental to the development of strong interpersonal relationships and helps counteract loneliness and isolation, which is becoming a growing public health challenge [15]. Mechanistically, it has been posited that ARHL communication challenges reduce active social participation; this withdrawal has a cascading effect leading to social isolation and feelings of loneliness; finally, there are also increased risks to health and mortality [16–19]. While these risk factors are associated with general ageing [20, 21], these risks are significantly exacerbated by hearing loss.

There is currently no way to reverse ARHL and management is generally focused on the provision of assistive listening devices such as hearing aids to amplify sounds. However, despite the noted benefits of hearing aids for communication and broad quality of life outcomes [22], hearing aid adoption rates are low, with global estimates indicating 17% of persons that could benefit from a hearing aid do so [23]. Hence, most older adults with hearing loss have *unaddressed* hearing loss [24], acknowledging this has been referred to synonymously as untreated [17], unperceived [25], or uncorrected [26]. To the best of our knowledge, there is no gold-standard terminology and there may be subtle variations in definition. For the purposes of our study, a participant with unaddressed hearing loss refers to an individual with a mild-to-moderate sensorineural hearing loss that does not currently use an assistive listening device such as a hearing aid.

## Benefits of group singing

The importance of social support, social networks, and social participation has been highlighted as a target for audiological rehabilitation, specifically for adults with ARHL [19]. In response to this, music and arts-based interventions for health and wellbeing have come to prominence in recent years with the benefits of being effective for preventing and promoting better health outcomes; helping to manage and treat health; and being engaging and cost-effective [27]. One potential non-clinical intervention showing promise is group singing—a universally practiced music activity that is more common than individual singing in approximately 70% of all societies [28]. There is an intimate and important cultural connection between music making and social context, with group singing being strongly associated with religious practice, dance, games, ceremony, and work [28].

Community choirs are popular, with 3.5 million Canadians having sung in 28,000 choirs in 2016 [29]. Typically consisting of amateur singers, the focus is to provide a supportive and inclusive environment where people can enjoy singing. These activities promote community engagement, cultural enrichment, and social connections among participants. Moreover, a diverse group of older adults that participated in a methodologically robust community choir intervention experienced significant benefits to loneliness and interest in life, compared to controls [30].

A potential benefit of group singing is the enhancement of speech perception, which is particularly relevant as ARHL is primarily associated with difficulties perceiving SIN [9–11]. Here, we provide a brief overview of two key theoretical frameworks that support speech perception: the OPERA hypothesis [31–33], and the Processing Rhythm in Speech and Music (PRISM) framework [34].

The OPERA hypothesis posits that music may engage speech processing networks if certain conditions are met. 1) *Overlap*—sensory and cognitive processing of music and speech is encoded by brain networks that overlap. More commonly referred to as neural sharing, there is ample neurological evidence to support the overlap between music and speech processing [35, 36]. 2) *Precision*—music must place a higher sensory and cognitive demand than speech. For example, pitch perception thresholds in music tend to operate at the semitone interval, whereas the speech intonation curve of a question utterance is much larger, typically extending beyond an octave (or 12 semitones) [37, 38]. 3) Music training that is enjoyable, regularly practiced, and engaging will satisfy the final requirement related to *Emotion*, *Repetition*, *and Attention*.

The PRISM framework suggests that three rhythm processes in both speech and music may lead to improved speech perception. 1) *Precise auditory processing*—both speech and music perception require accurate processing of small deviations. However, the regularity and

precision required in music production (particularly when performing with others) may fine-tune the auditory system, leading to enhancements in perceiving speech processes such as temporal envelope [39], or formant tracking (such as in plosives such as /ba/ and /da/) [40]. 2) *Synchronization/entrainment of neural oscillations to external rhythmic stimuli*—because musical rhythm typically features temporal regularities, music is a useful stimulus to entrain neural oscillations. Entrainment optimizes processes associated with attention and prediction [41], and hierarchical processing (i.e., phonemic, syllabic, phrase level speech perception) [42, 43], playing a critical role in speech and music perception. 3) *Sensorimotor coupling*—auditory and motor networks are linked and implicated in effective processing of both speech and music domains [44, 45]. Several studies have shown that music training has the capacity to strengthen sensorimotor coupling [46, 47], supporting a potential mechanism for the musician advantage.

Group singing also confers a unique opportunity to practice sensorimotor coupling—the dynamic integration of auditory, somatosensory, and motor systems [48, 49], which likely involves a dorsal pathway and its constituent fibre tracts integrating auditory cortices to the inferior frontal gyrus through the posterior parietal cortex [50]. This same dorsal pathway has been implicated in speech perception in noise, particularly as listening conditions become more challenging [51]. However, because a choir will experience tempo drift and micro-perturbations of intonation, a very high demand is placed on brain areas and fibre tracts subserving sensorimotor coupling. Singers in a choir are understandably motivated to coordinate sensory input and motor output to ensure accurate and harmonious vocal performance. Thus, motivated practice over a period of weeks and months may lead to neuroplastic changes (e.g., enhanced motor areas of the brain), which may ultimately support SIN perception [52, 53].

The musician advantage was brought to prominence in 2009 and is supported by empirical evidence that musicians outperformed non-musicians in SIN tasks—with years of consistent musical practice being positively associated with better SIN, working memory, and pitch perception [54]. More recently, a systematic review [55], and meta-analyses [56, 57], provided further support for this claim. Musicians routinely outperformed non-musicians on various SIN tasks, with more pronounced benefits across lower signal-to-noise ratios (SNRs), as well as across different distractor types and target properties, such as sentences, single words, phonemes, and tones. Other studies have shown stronger positive effects of musicianship on SIN performance across more challenging listening situations, suggesting greater benefits when the auditory system is under greater strain. One possible mechanism involves the type of training musicians receive (i.e., focusing on specific sounds in compositions that contain multiple simultaneous sounds), which may help them to better segregate auditory information in noisy environments. Neuroimaging studies have also revealed better auditory discrimination for musicians at the neural level (e.g. stronger spectral amplitudes in the Frequency-Following Response (FFR), more precise timing of transient response peaks, and improved Event-Related Potentials (ERP) waves such as P2 and N400). Studies have also shown a musician pattern of neural recruitment via shared pathways (both ventral and dorsal auditory streams in both hemispheres). The musician advantage in SIN may also be influenced by cognitive factors such as auditory working memory and selective attention, as well as age-related differences, with older musicians showing a greater SIN advantage compared to non-musicians of the same age. Additionally, an early age of onset and more extensive duration of musical training is correlated with stronger neural encoding and better performance on SIN tasks. Overall, these studies revealed a complex interaction between lower-level auditory processing and higher-level cognitive functions, indicating that musical training may enhance SIN perception through neuroplasticity and improved auditory processing. Despite these positive findings, some studies show variability in results, with either no significant advantage observed, or an advantage only under specific experimental conditions. A recent review raises questions

regarding the causality of music training and its transfer to non-musical benefits, such as SIN perception, are often weak or nonexistent [58].

Hence, the role of music training for individuals with hearing loss has been explored with cautious optimism, with the possibility that music training may enhance SIN perception as well as psychosocial outcomes. However, a systematic review of studies that tested the hypothesis that music training can enhance speech understanding in people with hearing loss showed that most of the studies had flaws in their methodologies and design such as no (or an inappropriate) control group, lack of randomisation, failure to account for multiple comparisons, or inadequate limiting of participant or tester bias [59]. Additionally, the four studies with appropriate designs did not show any benefit to speech understanding [60–63].

The strongest evidence that group singing enhances SIN for older adults with hearing loss, was a study involving a choir-singing group and an age- and audiometrically-matched control group, which did not participate in any activities, over a 10-week period [64]. Both groups were carefully matched based on previous musical experience, quantified by the years of formal musical training, thereby enabling causal inferences that are typically unachievable in correlational or cross-sectional studies of musicians' auditory capabilities. The choir participants exhibited significant improvements in SIN perception, pitch discrimination, and the robustness of the neural encoding of speech stimuli (frequency following response, FFR). The enhancement in SIN perception among choir participants was mediated by improved pitch discrimination, which was, in turn, predicted by the strength of the FFR. Although these results support the hypothesis that short-term participation in choir singing serves as an effective intervention for mitigating ARHL, reasonable concerns about participant bias may be raised due to the use of a self-selected choir group and a do-nothing control group.

Several systematic reviews have been conducted exploring the psychosocial benefits of singing (which is predominantly focused on group singing) in a range of populations and health conditions [65–68]. Overall, these findings indicate that group singing is an enriching activity that shows promise, but more research needs to be conducted to understand the therapeutic benefits for psychosocial wellbeing. Group singing was associated with better health-related quality of life, reduced mental distress, reduced symptoms of anxiety and depression, enhanced mood and emotional states, self-efficacy, a sense of purpose and achievement, social connectedness, and a sense of community and belonging. Physiological benefits have also been shown, with coordinated group singing leading to enhanced synchronization of respiration and heart rate variability (HRV) [69, 70].

There are several frameworks that suggest how and why music provides psychosocial benefits. Firstly, the musical and social bonding hypothesis posits that music-making is a coevolved system for social bonding [71]. The musical and social bonding hypothesis suggests that music can be likened to a toolkit, consisting of the following features: rhythm and dance (which supports synchronization and coordination); melody, harmony, and vocal learning (which supports unison and harmony); repetitive structure (which supports prediction); and music and social identity (which supports group identity). Importantly, these are all features common to group singing—or choir singing—which is a popular form of active music participation in the Western world. Notably, the musical and social bonding hypothesis argues that music making is more effective for large-scale bonding than other socially-facilitated contexts such as grooming or language [71].

Music has also been conceptualized in relation to its therapeutic benefits, consolidated into seven capacities in the Therapeutic Music Capacities Model [72]: 1) *Music is engaging*, utilizing a wide range of cognitive functions such as attention and working memory. There is evidence that long-term music training induces neuroplastic changes [73, 74]. Furthermore, while still inconclusive, there is some evidence that suggests music may enhance cognitive reserve [75],

and limit age-related cognitive decline [76, 77]; 2) *Music is emotional*, and can induce a wide variety of emotional states, and heighten arousal and reward [78]; 3) *Music is physical*, with movement and dance a typical component to music engagement. Significantly, physical activity (such as dance) is a known contributor for supporting healthy ageing and cognition [79, 80]; 4) *Music permits synchronization*. The ability of individuals to synchronize to an external rhythm (such as a musical beat) is a fairly unique feature of musical activity. Group synchronization is associated with cooperation, group cohesion, collective identity, enhanced non-verbal communication, and better learning outcomes [81–84]; 5) *Music is personal*, allowing for personal connection, such as the reinforcement or re-evaluation of self. This is particularly important in the context of ARHL, as the construction and negotiation of identity and self may face pressures due to age and hearing loss stigma [85, 86]; 6) *Music is social*, and helps create bonds and connections with greater efficacy compared to other group-based interventions, such as crafting [87]; and 7) *Music is persuasive*. The authors of the Therapeutic Music Capacities Model argue that music holds a unique cultural position in cultural contexts such as religion, advertising, and health, that may reinforce and support positive treatment outcomes. Thus, music has the potential to amplify therapeutic benefit, or enhance adherence to a program.

To summarise, in the context of a rapidly ageing population, untreated hearing loss has significant consequences that affect communication and psychosocial wellbeing at the individual and community level. On the other hand, there is promising but limited evidence that group singing may support both communication and psychosocial outcomes. However, numerous systematic reviews have consistently highlighted limitations that limit generalizability as well as the ability to formally recommend group singing as an intervention with robust psychosocial benefits [65–68]. These include heterogenous populations; small sample sizes; single site studies lacking control groups (or appropriate control groups); a lack of theoretical frameworks; a lack of large and well-designed experimental randomized controlled trials (RCTs); and a level of bias that was typically moderate to high [65–68]. Hence, to address these concerns, this SingWell Project [88] study utilizes the theoretical frameworks of the musical and social bonding hypothesis, Therapeutic Music Capacities Model, OPERA, and PRISM that will guide hypothesis testing; adopted a multisite design to recruit a sufficiently large sample size based on a statistical power analysis; will limit of risk of bias through our recruitment strategy and experimenter blinding; and will use an RCT design with planned statistical analyses.

## Aims and hypotheses

The aim of this study is to explore if group singing is beneficial for older adults with unaddressed hearing loss, hypothesizing that group singing may be effective at improving speech perception and psychosocial outcomes. At the macro-level (i.e., across a 12-week period), the outcomes of interest include: 1) speech-in-noise perception, and 2) psychosocial health. At the micro-level (i.e., pre- and post-session effects at week 2, 7, and 11), the outcomes of interest are 1) psychosocial health, and 2) heart rate variability. Note that additional detail for macro and micro-level tests, and the active control group is described in the Materials and Methods section.

## Macro-level hypotheses

H1. Group singing may be associated with better music perception such as pitch, rhythm, timbre, and higher-level music skills—more so than participants in the control group.

H2. Group singing may be associated with better speech perception such as speech-in-noise and emotional prosody—more so than participants in the control group.

a. The mechanism for speech perception enhancement may be due to improvements in music perception (i.e., pitch, rhythm, timbre, or higher-level music skills.)

H3. Group singing may be associated with better psychosocial health—more so than participants in the control group.

a. The primary outcome of interest is the enhancement of quality of life and social connectedness.

b. The secondary outcomes of interest are the reduced symptoms of depression and anxiety, and the enhancement of self-esteem.

### Micro-level hypotheses

H1. Group singing may lead to increases in positive mood—more so than the participants in the control group.

H2. Group singing may foster greater feelings of social closeness—more so than the participants in the control group.

**Exploratory analyses.**   Finally, in an exploratory manner, we aim to explore the role of HRV and investigate the role of individual differences—for example, are the benefits of group singing specific to participants with poorer baseline outcomes? These findings may help us understand the specific populations that may benefit from group singing, to optimize future interventions.

## Materials and methods

### Ethics

Ethical approval to conduct this study was obtained from the Toronto Metropolitan Research Ethics Board (ID: REB 2024–103). A copy of the ethics protocol is available in S1 Appendix. All participants will be required to provide informed written consent.

### Study design

A schematic of the study design can be seen in Fig 1. The study is a longitudinal RCT. Initially, participants will undergo an eligibility screening process and complete an information and consent form (described in detail under 'Testing Procedure'). Participants will be randomly assigned to either group singing or audiobook club (control group) intervention for a training period of 12-weeks and will be unaware if they are in the experimental or control group (i.e., participant-blinded). After this, the study has multiple timepoints for testing, that are broadly categorized as macro, or micro timepoints.

The rationale for the use of an audiobook club as a suitable control group is due to several factors. Firstly, listening to audiobooks is a common clinical recommendation for hearing loss and have been used in other RCTs as a control [89, 90]. Secondly, discussion from the audiobook club will generate asynchronous speech, which is a suitable compliment to investigate the targeted benefits of synchronized group singing. Finally, we anticipate that the credibility and expectation between both interventions will be similar given they are both group activities in the auditory domain.

This study was registered as a clinical trial (clinicaltrials.gov, U.S. National Library of Medicine, ID: NCT06580847); and adheres to a Standard Protocol Items: Recommendations for

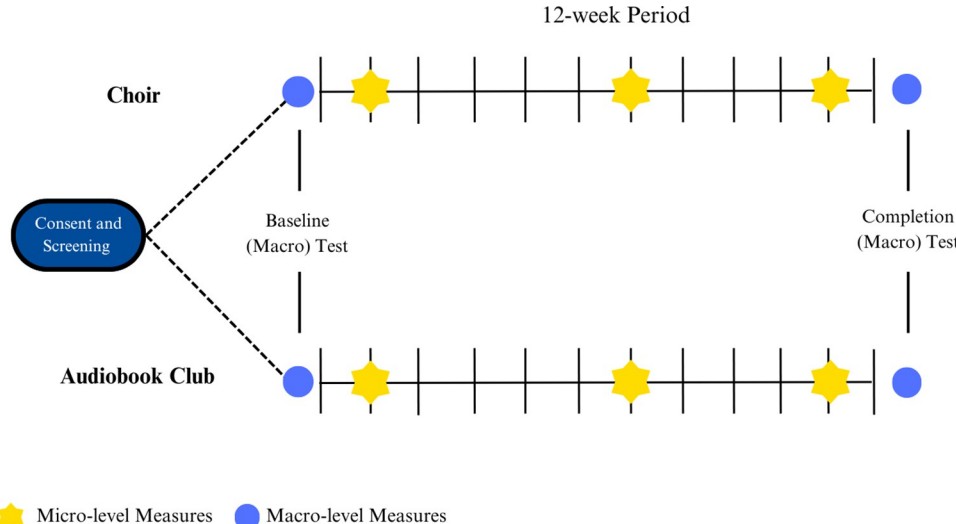

Fig 1. Overview of the study design.

Interventional Trials (SPIRIT) Checklist with 2022 extension [91–93], which is provided in S2 Appendix. Any modifications to the protocol will require approval from the Toronto Metropolitan Research Ethics Board and will be noted in the subsequent registered report article.

## Participants

Participants will be recruited from seven sites around the world: Toronto, Montreal, and St John's (Canada); Los Angeles (United States of America); Groningen (Netherlands); Oldenburg (Germany); and Adelaide (Australia). Each site will aim to recruit a total of 30 participants with a simple randomisation using a 1:1 computer generated allocation (15 randomly assigned to group singing, and 15 randomly assigned to the audiobook club), for a total of 210 participants among all the sites. Participants will not be able to change their allocation.

Recruitment will be variable as each site will have access to different databases, networks, and communities. Hence, recruitment flyers will be sent to a range of organisations such as: deaf and hard-of-hearing organisations, aged-care homes/retirement centres, senior centres, veterans organisations; and to databases such as the Toronto Metropolitan University Auditory Participant Pool and the Toronto Metro Senior Participant Pool. To avoid potential participant bias, flyers will advertise that the study is looking for participants to engage in a creative group activity, rather than group singing or an audiobook club, specifically.

**Inclusion criteria.** 1) Adults aged 60 years and older; 2) Bilateral mild-to-moderate hearing loss (20–49 dB hearing level, as proposed by the Global Burden of Disease Expert Group [94]), measured using four-frequency pure-tone average across both ears (4FPTA) measured at 500 Hz, 1000 Hz, 2000 Hz, and 4000 Hz; 3) Unaddressed hearing loss (i.e., participants must not currently use a hearing aid, cochlear implant, or assistive listening device); 4) No significant cognitive impairment, to be assessed with the Montreal Cognitive Assessment for people with hearing impairment (MoCA-H) [95], with participants requiring a score ≥ 24; 5) Not use a pacemaker or anti-arrhythmic agents/medications; 6) Not currently participating in regular active music learning (e.g., choir, formal music training) or audiobook clubs within the last year; and 7) Sufficient language capacity to understand and complete the test materials. Note: all materials will be presented written and/or aurally in English at the sites located in Canada, United States of America, and Australia; Dutch at the Netherlands site; and German at the

Germany site. Test material examples corresponding to English, Dutch, and German will be provided where applicable.

### Test battery

**Cognitive screening.** MoCA is a validated brief screening tool designed to detect mild cognitive impairment [96]. MoCA-H is an extension of the MoCA, designed specifically for adults with acquired hearing loss [95]. The cognitive domains assessed include short-term memory recall, visuospatial abilities, executive functioning, phonemic fluency, verbal abstraction, attention, concentration, and orientation.

**Demographics, social networks, musical background, credibility, and expectation.** Participants will be asked to provide their demographic details such as age, sex, gender, residential status, and education. We will also measure their social network diversity (e.g., connections with friend, families, neighbours, etc.) with a subscale from the Social Network Index [97]. Participant's previous musical training will be measured using the Musical Training subscale from the Goldsmith's Musical Sophistication Index (Gold-MSI) [98]. This 7-item questionnaire captures the participants' history of formal musical training with more points representing more musical training. Participant perspectives on the credibility and expectations in respect to their assigned group will also be recorded with the credibility/expectancy questionnaire [99].

**Speech perception materials.** The Coordinate Response Matrix (CRM) sentence recognition test will be used to evaluate speech-in-noise (SIN) perception. Our CRM test is based on the original versions that were developed in English by [100–103]. The carrier phrase is a sentence with a call sign of cat or dog and where one of six colors and one of eight numbers are mentioned (e.g., "Show the dog where the green (color) three (number) is."; "Laat de hond zien waar de groene (color) dire (number)" [Dutch]; "Zeige dem Hund, wo die grüne (color) Drei (number) ist" [German]. We use a two-down/one-down adaptive procedure based on correct color and number identification. Initial step sizes are 10 dB, which is set to 3 dB after two reversals. The test ends after an additional four reversals, generating a speech reception threshold (SRT).

EmoHI is a vocal emotion perception test [104]. The stimuli consist of pseudospeech sentences such as 'Koun se mina lod belam' and 'Nekal ibam soud molen' that are not meaningful in any Indo-European languages and are based on the Geneva Multimodal Emotion Portrayal (GEMEP) Corpus materials [105]. Stimuli were spoken by four monolingual native Dutch speakers (two male, two female) without any discernible regional accent, consisting of three emotions: happy, angry, and sad. These were selected for the widest applicability across age ranges [106], and for maximum comparability of the EmoHI test across populations of differing hearing profiles [104]. Stimuli were recorded in an anechoic room at a sampling rate of 44.1 kHz and equalized in RMS. The materials and methods of the EmoHI test are the same as in a previous study [104], but the current interface will not have a child-directed game-like environment. Participants will instead be instructed to listen to the pseudo-sentences and determine whether the voice of the speaker sounded happy, sad, or angry, by clicking on one of three corresponding buttons on the screen labeled "blij"/happy, "verdrietig"/sad, "boos"/angry. For practice, three additional stimuli (one per emotion, one production per speaker) will be used for the training session, but not included in the experimental phase. For the experiment, 36 vocal emotion stimuli are used, comprising the categories happy, sad and angry, with 12 items in each category (4 speakers x 3 utterances). Each emotion category comprises both arousal and valence dimensions: either high or low arousal, or positive or negative valence. Practice items, but not experimental items, will be presented with feedback.

Experiment items will be presented in randomized order in one block of approximately 6 to 8 minutes.

**Music perception materials.** A Frequency Difference Limen (FDL) task will be used to evaluate pitch perception. Participants will be presented with a three-alternative forced choice (3AFC) paradigm containing two pure tones (200 ms in duration, with 20 ms envelope rise and fall times) at 500 Hz, with the target stimulus adaptively presented at a higher frequency. The participant will identify which corresponding tone is higher on a computer keyboard (i.e., the first, second, or third tone). A pitch discrimination threshold will be calculated with an adaptive staircase procedure with the frequency difference between the target and 500 Hz tones divided by two after three correct responses, or multiplied by two after one incorrect response. After five reversals, the step size will change, with the frequency difference divided by 1.414 after 3 correct responses or multiplied by 1.414 after one incorrect response. Participants will complete two-blocks, with each block ending after 12 reversals. The calculation of their FDL threshold is determined from the mean of the last 10 reversals of each block, which were then averaged between the two blocks.

The Beat Alignment Test (BAT) will be used to measure musical beat perception and rhythm [107]. The BAT consists of 36 musical excerpts that are played with a superimposed click track in three conditions that are: 1) "On beat"—on the beat; 2) "Off beat"—off the beat with the wrong tempo; and 3) "Phase error"—out of phase. Participants will be provided with practice trials. The task is to identify if the excerpt is on the beat by responding on a keyboard with YES or NO. Participants will be instructed not to tap or move along to the music.

The Spectral-temporally Modulated Ripple Test (SMRT) is a measure of spectral resolution [108], which will be used as a proxy measure of timbre perception. Compared to typical spectral resolution tasks, the SMRT features dynamically changing ripples. Participants will be presented with a three-alternative forced choice (3-AFC) task. The target choice is presented at an initial presentation of 0.5 ripples per octave (rpo), while the other two choices are presented at a reference stimuli of 20 ripples rpo. The target stimuli are modified with a 1-up 1-down adaptive procedure with a step size of 0.2 rpo. After 10 reversals, a threshold is derived from the average of the last six reversals.

The Music-in-Noise Task (MINT) will be used to measure inter-individual differences in hearing-in-noise skills [109]. MINT is utilized as a measure of higher-level musical skills. The MINT uses a match-mismatch trial design, whereby participants first listen to a short instrumental excerpt presented in "multi-music" noise, followed by either a matched or scrambled version of the excerpt presented in silence, and are asked to judge whether the two excerpts are the same. The MINT includes five listening conditions, of which we will utilize three (Baseline, Rhythm, and Prediction [reversed order]), which differ according to the presence or absence of different types of contextual cues, which have been shown to be differentially sensitive to musical and linguistic expertise [109]. The Baseline condition provides no additional cues. In the Rhythm condition, the tones of the instrumental excerpts are all the same pitch within each trial. As such, rhythmic variation between matched and mismatched pairs represents a form of rhythmic cueing. In the Prediction condition, the excerpts are presented in silence first, followed by the noise-masked pair. This predictive cueing helps to anticipate incoming information when the excerpt is presented in noise. Each MINT condition includes 20 trials, with four familiarization trials for each condition prior to the start of the main task. Difficulty is also varied by adjusting the signal-to noise ratio (SNR) between the musical excerpt and the background noise, such that musical excerpts will be more or less hidden in the background noise. Conditions will be tested at four different signal-to-noise levels (0, −3, −6, and −9 dB).

**Psychosocial questionnaires.** *Macro-level* q*uestionnaires.* The 12-item short-form survey version 2 (SF-12v2) from the Medical Outcomes Study (MOS) is a widely used health-related

quality of life questionnaire that measures eight health concepts: 1) Physical functioning; 2) Role physical; 3) Bodily pain; 4) General health; 5) Vitality; 6) Social functioning; 7) Role emotional; and 8) Mental health; aggregated as a Physical Summary and a Mental Summary [110]. We will utilize the 1-week (acute) recall period. Scores range from 0 to 100, where a zero score indicates the lowest level of health and 100 indicates the highest level of health.

The Hospital Anxiety and Depression Scale (HADS) consists of 14-items designed to measure symptoms of psychological distress in medical patients [111]. HADS consists of two scales, one for anxiety (HADS-A) and depression (HADS-D). It is an efficient screening instrument that is commonly used in clinical contexts to identify and quantify symptoms related to anxiety and depression. While the original version was validated with adults aged 18 to 65 years; HADS has also been validated as an assessment for adults aged 65 to 80 years old [112]. Scoring for each item ranges from 0 to 3, with a higher score corresponding to higher levels of anxiety or depression.

The Collective Self-Esteem Scale (CSES) is a 16-item scale that evaluates individual and collective identities [113]. There are four distinct subscales (hence there is no total scale score) measuring: 1) membership self-esteem; 2) private collective self-esteem; 3) public collective self-esteem; and 4) and importance to identity, on a scale between 1 and 7, with higher scores associated with higher levels of self-esteem. The CSES is widely used with good reliability.

The revised Social Connectedness Scale (SCS-R) consists of 20-items that measures the psychological sense of belonging and interpersonal closeness in social contexts [113]. Reponses are measured on a Likert scale ranging from 1 = strongly disagree to 6 = strongly agree on items that represent both positive and negative aspects of social connectedness.

*Micro-level questionnaires*. The Positive and Negative Affect Schedule (PANAS) is a widely used measure of mood or emotion that consists of 20-items (10 positive and 10 negative affect items) [114]. Participants will be asked to "Indicate to what extent you feel this way right now, that is, at the present moment" on a scale of 1 = very slightly or not at all; 2 = a little; 3 = moderately; 4 = quite a bit; and 5 = extremely; in respect to affective words such as "interested" or "afraid".

The Inclusion of Other in the Self Scale (IOS) is a single item that measures how close an individual feels with another person or group [115]. The tool contains 7-pairs of circles consisting of the word "Self" in one circle, and the word "Community" in the other circle. Each pair of circles differs in proximity, from 1 = no overlap; 2 = little overlap; 3 = some overlap; 4 = equal overlap; 5 = strong overlap; 6 = very strong overlap; 7 = most overlap. Participants will select the image that best describes their relationship with their randomly assigned group.

The Subjective Units of Discomfort Scale (SUDS) consists of a 0 to 100-point scale measuring self-rated anxiety and discomfort [116]. We will utilize descriptive anchor points at 0 = "no anxiety, calm, relaxed"; 25 = "mild anxiety, alert, able to cope"; 50 = "moderate anxiety, some trouble concentrating"; 75 = "severe anxiety, thoughts of leaving"; and 100 = "very severe anxiety, worst ever experienced".

**Heart rate variability.** Heart Rate Variability (HRV) will be measured using photoplethysmography (PPG) sampled at 135 Hz using Polar Verity Sense heart rate sensors. Each participant in the session will receive their own sensor and all sensors will be time synchronized. The measurement period will include the entire group-singing or audiobook club session. Additionally, we will measure pre- and post-singing/audiobook rest periods, each lasting 60 seconds as a baseline measure. PPG data will be imported to Kubios HRV Scientific software (version 4.1.1) which will allow for computation of Root Mean Square of Successive Differences (RMSSD) between adjacent peaks, a time-domain measurement of HRV. Synchrony of HRV between participants over the course of a session (i.e., between the two rest periods) will be assessed using time-frequency coherence analysis [117]. A pre-post comparison of

HRV during rest periods will also be conducted for each participant to assess global changes in ANS activity over the course of a single session.

## Test procedure

Below, we list the three facets to the test procedure, noting that some occur over several sessions. For consistency and logistical pragmatics, we will aim to test all participants within two-week windows, relative to key training timepoints.

1. Consent and Screening—initial session.

2. Macro Timepoint Testing—two sessions in a test booth:

   a. Baseline (up to 2-weeks before training begins).

   b. Completion (up to 2-weeks after training ends).

3. Micro Timepoint Testing—six sessions at the site of training:

   a. Pre- and Post-session testing at Week 2.

   b. Pre- and Post-session testing at Week 7.

   c. Pre- and Post-session testing at Week 11.

**Consent, screening, and demographic information.** Data collectors will assess participant's eligibility. This will consist of understanding and signing a participant information and consent form; completion of an audiological assessment to establish their level of unaided hearing loss (pure tone average testing at 500 Hz, 1000 Hz, 2000 Hz, and 4000 Hz); and passing the MoCA-H screener with a criterion score $\geq$ 24. If these inclusion criteria are met, participants will complete their demographic information in a questionnaire.

**Macro timepoint testing.** The purpose of the testing from *baseline* to *completion* is to examine any benefit from the training. Macro timepoint testing consists of the following behavioural measures presented in fixed order: CRM, EmoHI, FDL, BAT, SMRT, and MINT; and the following questionnaires: SF-12v2, HADS, CSES, SCS-R, EMO-CheQ.

The macro timepoint testing will occur on-site in an acoustically-treated test booth. The presentation level of behavioural measures will be calibrated to 65 dB with a sound-level meter measured at the participants' position, which will be located 0.5 m directly in front of the loudspeaker. Data collectors conducting the macro timepoint tests will be blinded to the participants' intervention and participants will be asked not to talk about their intervention during testing. Any instances of unintentional blinding (i.e., a participant informing the data collector of their participation) will be recorded. The participants will receive $20 CAD (or equivalent currency) for the consent and screening, and each macro timepoint testing session.

**Micro timepoint testing.** The purpose of the micro timepoint testing is two-fold. Firstly, to examine the physiological and psychosocial changes that occur from *pre-* to *post-session*; and secondly, to examine the cumulative longitudinal impacts on physiological measures for each intervention. Micro timepoint testing consists of HRV and the following questionnaires presented in fixed order: PANAS, IOS, SUDS.

The micro timepoint testing will occur on-site where the training occurs at: 1) Pre-session (directly before the intervention begins); and 2) Post-session (directly after intervention ends). Testing will involve biophysical measures and questionnaires (described in detail under the Test Battery and Test Procedure sections). These micro measures will be taken at the training

sessions in Week 2, 7, and 11 and designed to be indicative of pre- and post-session measures corresponding to the start, middle, and end of the training program.

## Training procedure

Participants will be randomly assigned to either the group singing or audiobook club intervention. Training will occur once-a-week over a total duration of 12-weeks, with each training session lasting approximately 1.5-hours, including informal networking and discussion. Adherence to the intervention will be reported.

Group singing will be facilitated by a choir master with at least one-year of experience leading a choir, a piano accompanist, and support from up to two research assistants who will sing with the participants. The broad goal of the group singing is to provide synchronous vocal pitch training in a supportive and casual environment. The beginning of each session will involve approximately 15-minutes of warm-up exercises, and the repertoire will be decided through discussion between the choir master and participants. Given the choir master will have some flexibility and need to adapt the sessions depending on group ability, a full outline of each lesson will be provided as supplementary materials in the final report. Participants will also be assigned an individual online 'homework' task each week—one-hour of using pitch-based games in Theta Music Trainer (https://trainer.thetamusic.com), an application designed for ear training and music theory [118]. The goal of the homework task is to maximize pitch-based perception. Theta Music Trainer provides logging capability that will allow us to monitor participants' weekly homework tasks. Email reminders will be sent to encourage participant adherence.

Participants in the audiobook club will listen to a 1-hour length audiobook segment prior to each audiobook club session. Participants will be provided with a pool of 10 audiobooks with a brief description on genre and plot. Each audiobook club will collectively select which audiobook/s they would like to listen to during their 12-week intervention. Given they are expected to listen to ~1-hour per week; the audiobook/s must total approximately 12-hours in total duration. The pool has been selected to cover a wide range of genres and interests, and are available in English, German, and Dutch. These sessions will feature a facilitator that will constructively guide the participants through a series of open-ended questions developed by the research team that has been designed to elicit open and constructive discussion. Up to two research assistants will also be present and contribute to the discussions. Email reminders will be sent to encourage participant adherence to the weekly audiobook segment.

## Dissemination plan

The main findings will be published in the subsequent registered report article. Given the multisite nature of this study, individual sites may choose to develop their own specific hypotheses that extend beyond this protocol, and may choose to develop additional outputs beyond the associated registered report article. The SingWell Project is also committed to knowledge mobilization activities involving key stakeholders such as choirs, physicians, hearing health professionals, and community centres. Full details can be found on their website (https://www.singwell.ca/) and from the SingWell Project protocol [88].

## Data analysis

**Statistical power analysis.** An a priori power analysis was conducted to determine the sample size needed to detect the smallest effect size of interest. We determined the number of participants needed to detect the cross-level interaction of group (i.e., group singing vs. audiobook club) and time (i.e., *baseline* vs. *completion*) on SIN perception. The power analysis was

conducted using traditional approaches where the resulting sample size refers to the level 2 clusters–in this case, the number of participants.

*G\*Power* version 3.1.9.4 was used to conduct the analysis for an ANOVA within-between factor interaction [119]. For a conservative estimate, we assumed a small effect size of, $\eta^2$ = .01, corresponding to an effect size of f = 0.1. We estimated the correlation among the repeated measures to be *r* = .62, based on a previous group-singing controlled trial that reported an intraclass correlation (ICC) of .62 [64]. Accounting for a 20% attrition rate [64], the power analysis (f = .10, $\alpha$ = .05, *r* = .62) suggests a total sample size of *N* = 206 participants, approximately *n* = 30 per site for 85% power.

In addition to the power analysis, we also calculated the reliability of our repeated measurements to assess for sufficient reliability for each participant based on [120] (1):

$$n_{min} = \frac{\rho(1 - ICC)}{(1 - \rho)(ICC)} \tag{1}$$

where *n* min is the number of time points and $\rho$ is the population correlation across all the timepoints. Given our two timepoints and estimated ICC of .62, we have adequate-to-good reliability of our timepoints, $\rho$ = .77.

## Planned statistical analyses

**Tests of equivalence.** Given the sample size and random assignment to group, we assume that covariates such as age, gender, education, musical training/background, credibility, and expectation will be evenly distributed across the participants. Thus, we expect equivalence between participants and their randomly assigned intervention. To test this assumption, we will use a well-known technique to test for equivalence, the Bayesian *t*-test [121, 122]. Each Bayesian *t*-test will produce a coefficient called the Bayes Factor (BF), a measure of the likelihood of the alternative hypothesis (i.e., that the two samples differ). BF of 1 or less, will be interpreted as strong evidence for the null hypothesis [123]. Upon discovery of a BF larger than 1 we will conclude that the groups are not equivalent on this variable, and this variable will be used as a covariant in subsequently modeling of measures of interest.

**Difference score and outlier rejection.** Outliers due to technical errors will be removed. Univariate outliers on continuous independent and dependent variables will be rejected based on a boxplot method, wherein a data point is an outlier if it falls 1.5 times outside of the interquartile range. Because each group may have unique distributions, we plan to reject outliers based for each group individually. For each of the micro-level dependent measures, a difference score will be computed (i.e., *pre-* to *post-session*).

**Modelling and inferential statistics.** Due to the nature of this research design, there are various levels at which the data will likely show high levels of covariance (e.g., participant, group and site level), also known as clusters. Not accounting for this covariance can lead to inaccurate estimates of standard error, leading to inaccurate hypothesis testing (Mass & Cox, 2004). A well-known modeling technique that can account for this is mixed-effects linear modeling (MLM) and is also well suited to handle missing data in case of attrition or outliers (Pinheiro & Bates, 2000). MLM accounts for covariance at various levels using random effects, which allow for variance in the slope, or intercept. These random effects are defined within the model parameters based on the clustering in the research design. For our purposes, we will define a random-intercept for individual participant, to account for covariance in the repeated measures, and a random-slope for group, to account for the group level covariance.

The third level of nesting (i.e., site) is not accounted for in the model because we do not have hypotheses relating to the different sites. Modeling the third level will result in an

unnecessary overly complicated model, but ignoring the third level of clustering could result in heteroscedasticity of model residuals, underestimated standard errors and biased *p*-values. To address this, we will plot and visually assess the proposed model's residuals. If there is heteroscedasticity of the model residuals, we will apply a robust standard error estimator to account for the clustering [124]. All modeling will be done in the *lme4* package [125] in R.

For each micro-level dependent variable, three separate MLMs will be defined: 1) an intercept-only model; 2) a main-effect only model ($DV_{change} \sim group + time$) and 3) an interaction ($DV_{change} \sim group * time$) model. For micro-level models, the time variable will reflect the time point of the micro-level observation and will allow us to model changes between the group across time. Similarly, three separate MLMs will be defined for macro-level dependent variables: 1) an intercept-only model; 2) a main-effect only model ($DV \sim group + session$) and 3) an interaction ($DV \sim group * session$) model. For both macro and micro-level models we will use a hierarchical modeling technique to determine which model is the best fit to the data. Model fit parameters will be compared between the three models and results of the model with the best fit will be reported on. Upon the discovery of a significant interaction, a follow-up simple slope analysis will be used to examine the differences in time (e.g., *baseline* vs. *completion*) across groups. For all hypothesis tests, alpha values will be set at 0.05.

**Mechanisms and contributions to speech perception.** If a statistically significant benefit from group singing is found for speech perception, we will examine if changes in musical perception (i.e., FDL, BAT, and SMRT) are potential mechanisms or contributors for this change in speech perception (i.e., CRM and EmoHI). To do this, a hierarchical method will be used to compare a model which predicts speech perception difference score from BAT, FDL, and SMRT difference scores to a model which also contains an interaction term between speech perception variables and the group variable. The model which is a better fit for the SIN change scores will be reported on, and any significant interactions will be decomposed using simple slope analysis.

## Supporting information

**S1 Appendix. Ethics protocol.**
(PDF)

**S2 Appendix. SPIRIT checklist.**
(PDF)

## Acknowledgments

We thank the following individuals and organizations for providing support and advice for this project: Laura Rachman, Kai Siedenburg, Andrew Oxenham, Kay Wright-Whyte, Katherine Spencer, Dilara Karabatak, Bridget Boyle, Rhiannon Ueberholz, Gurjit Singh, Sonova AB, Gaurang Khatri and Theta Music Trainer.

## Author Contributions

**Conceptualization:** Chi Yhun Lo, Benjamin Rich Zendel, Arla Good, Frank A. Russo.

**Formal analysis:** Carmen Dang, Sean Gilmore.

**Funding acquisition:** Frank A. Russo.

**Methodology:** Chi Yhun Lo, Benjamin Rich Zendel, Deniz Baskent, Emily Coffey, Assal Habibi, Ellie Harding, Gunter Kreutz, Mridula Sharma, Carmen Dang, Sean Gilmore, Helen Henshaw, Colette M. McKay, Arla Good, Frank A. Russo.

**Project administration:** Chi Yhun Lo.

**Supervision:** Chi Yhun Lo, Frank A. Russo.

**Writing – original draft:** Chi Yhun Lo.

**Writing – review & editing:** Benjamin Rich Zendel, Deniz Baskent, Christian Boyle, Emily Coffey, Nathan Gagne, Assal Habibi, Ellie Harding, Merel Keijzer, Gunter Kreutz, Bert Maat, Eva Schurig, Mridula Sharma, Carmen Dang, Sean Gilmore, Helen Henshaw, Colette M. McKay, Arla Good, Frank A. Russo.

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
