## [Decision Letter · Decision Letter 0]

4 Oct 2024

PONE-D-24-30955Speech-in-noise, psychosocial, and heart rate variability outcomes of group singing or audiobook club interventions for older adults with unaddressed hearing loss: a SingWell Project multisite, randomized controlled trial, registered report protocolPLOS ONE

Dear Dr. Lo,

Thank you for submitting your manuscript to PLOS ONE. After careful consideration, we feel that it has merit but does not fully meet PLOS ONE’s publication criteria as it currently stands. Therefore, we invite you to submit a revised version of the manuscript that addresses the points raised during the review process.

We look forward to receiving your revised manuscript.

Kind regards,

Michael Döllinger, Ph.D.

Academic Editor

PLOS ONE

Journal Requirements:

“Funding for this study is provided through a Social Sciences and Humanities Research Council of Canada (SSHRC) Partnership Grant awarded to F. Russo (Reference Number: 895-2021-1018)”

5. In your cover letter, please confirm that the research you have described in your manuscript, including participant recruitment, data collection, modification, or processing, has not started and will not start until after your paper has been accepted to the journal (assuming data need to be collected or participants recruited specifically for your study). In order to proceed with your submission, you must provide confirmation.

6. Your abstract cannot contain citations. Please only include citations in the body text of the manuscript, and ensure that they remain in ascending numerical order on first mention.

7. We note that the original protocol that you have uploaded as a Supporting Information file contains an institutional logo. As this logo is likely copyrighted, we ask that you please remove it from this file and upload an updated version upon resubmission.

Reviewers' comments:

Reviewer's Responses to Questions

**Comments to the Author**

1. Does the manuscript provide a valid rationale for the proposed study, with clearly identified and justified research questions?

Reviewer #1: Yes

Reviewer #2: Yes

Reviewer #3: Yes

2. Is the protocol technically sound and planned in a manner that will lead to a meaningful outcome and allow testing the stated hypotheses?

Reviewer #1: Yes

Reviewer #2: Yes

Reviewer #3: Yes

3. Is the methodology feasible and described in sufficient detail to allow the work to be replicable?

Reviewer #1: Yes

Reviewer #2: Yes

Reviewer #3: Yes

4. Have the authors described where all data underlying the findings will be made available when the study is complete?

Reviewer #1: Yes

Reviewer #2: Yes

Reviewer #3: Yes

5. Is the manuscript presented in an intelligible fashion and written in standard English?

Reviewer #1: Yes

Reviewer #2: Yes

Reviewer #3: Yes

6. Review Comments to the Author

You may also provide optional suggestions and comments to authors that they might find helpful in planning their study.

Reviewer #1: Important note: This review pertains only to ‘statistical aspects’ of the study and so ‘clinical aspects’ [like medical importance, relevance of the study, ‘clinical significance and implication(s)’ of the whole study, etc.] are to be evaluated [should be assessed] separately/independently. Further please note that any ‘statistical review’ is generally done under the assumption that study specific methodological [as well as execution] issues are perfectly taken care of by the investigator(s). This review is not an exception to that and so does not cover clinical aspects {however, seldom comments are made only if those issues are intimately / scientifically related & intermingle with ‘statistical aspects’ of the study}. Agreed that ‘statistical methods’ are used as just tools here, however, they are vital part of methodology [and so should be given due importance]. I look at the manuscript in/with statistical view point, other reviewer(s) look(s) at it with different angle so that in totality the review is very comprehensive. However, there should be efforts from authors side to improve (may be by taking clues from reviewer’s comments). Therefore, please do not limit the revision only (with respect) to comments made here.

COMMENTS: Because the study & manuscript is excellent in all aspects, there is hardly anything to point-out. However, I have different opinion regarding a few (only two) issues which are given below:

When lines 599-602 it is stated that “Given the sample size and random assignment to group, we assume that covariates such as age, gender, education, musical training/background, credibility, and expectation will be evenly distributed across the participants. Thus, we expect equivalence between participants and their randomly assigned intervention” then why further said “To test this assumption, we will use a well-known technique to test for equivalence, the Bayesian t-test”? Test proposed is appropriate and reference quoted [Stefan AM, Gronau QF, Schönbrodt FD, Wagenmakers EJ. A tutorial on Bayes Factor Design Analysis using an informed prior. Behav Res Methods. 2019;51: 1042–1058] is nice. However, it is often very rightly advised that when random allocation/assignment is used/done any statistical comparison of baseline characteristics is not necessary. In this context, I request authors to note that even if P-value(s) turn(s) out to be significant (while comparing baseline characteristics despite random allocation), it is, by definition, a false positive as you then are supposed to be testing ‘randomization’ then, which in any single trial may not balance all baseline characteristics {though I am sure that these learned authors already know these things}.

Please note that any regression techniques are not basically/originally developed for any sort of [between or within group(s)] comparison(s). Using mixed-model regression is very appropriate in given situation, but note that ‘Head-to-head’ comparison is expected, as this is an indirect/secondary/by-product testing, in my opinion.

As pointed out in ‘important note’ above “This review pertains only to ‘statistical aspects’ of the study and so ‘clinical aspects’ should be assessed separately/independently. In my opinion, to make this article acceptable (which is quite possible and easy), a small amount of re-vision (re-drafting) may be needed. ‘Minor revision’ is recommended.

Reviewer #2: The authors present a Registered Report Protocol on a randomized and controlled study on the effect of group singing on speech perception in noisy surroundings and psychosocial health in a cohort comprising 210 participants (including controls).

The research direction, the paradigms used, and the analysis presented are excellent. The authors gave a comprehensive overview in the introduction section, considering plenty of topics and references recently published and motivating their project. Also, the method section is comprehensively written and seems reproducible.

Furthermore, this study generally meets the criteria to be considered for publication in Plos One. Nevertheless, some points should be addressed before being considered for publication.

General

• Please avoid some acronyms/abbreviations, such as HA, MSB, TMCM, etc., that were only used in a particular paragraph or section. Using these abbreviations makes it hard to read the text and is not necessarily needed.

• Fig. 2 shows no additional relevant information and should be eliminated.

• I miss a paragraph on „Expected results“. The statements in l.258-265 are more general. Can you give numbers on expected outcomes?

Introduction

• l.252: Please replace MSP with MSB (but see my comment on abbreviations before)

Methods and Methods

• l. 376: hund -> Hund

• l.376: us -> used

• l.339 & l.521: Hearing loss-related inclusion criteria will be evaluated at four frequencies of 500, 1000, 2000, and 4000 Hz, but audiological assessment at 1000, 2000, 4000, and 8000 Hz. Why this discrepancy? This discrepancy is especially relevant for older patients, where hearing loss affects higher frequencies.

Data Analysis

• l.609/610: Please carefully check the syntax of Eq. (1). I suppose „min“ should be subscripted.

• l.671-673: Please avoid R-syntax. It would be more helpful to explain what will be dependent variables and which explanatory variables (and their interaction) will be considered.

Reviewer #3: This is a thorough and well-planned study regarding a promising subject.

ARHL affects large parts of the population and brings with it further serious life limitations on social, emotional and cognitive levels. Group singing as culturally anchored activity could be a simple and inexpensive method to counteract the negative consequences of ARHL, and there is already some evidence of effectiveness. However, previous studies have flaws that do not allow any watertight conclusions to be drawn. The presented study attempts to generate clarity through a large number of test subjects and control groups, the multi-site approach and extensive multi-level testing and statistics. The study meets the PLOS ONE publication criteria.

I have only a few remarks listed below.

General: Regarding the central point of the study, I would appreciate a little more detailed outline of the group singing sessions. It is totally legitimate that there can (must!) be flexibility and adjustments in the sessions’ progressions and I acknowledge that each protocol will be supplemented in the final report. However, since there are many different approaches for choir/group singing, it would be interesting to know a few conceptual details, e.g. whether there is going to be a warm-up of which duration and content or what repertoire is roughly in focus. What are the choir leaders’ qualifications/specifications?

Title: The title seems very long – is there an possibility to shorten it?

L. 31: The word „appropriate“ is somewhat undefined: Could you insert something that makes clear what it means in this context?

L. 52: „Older adults“ seems undefined. I would appreciate if you gave their actual age range, where possible, i.e., 60+

L. 61: Is there a conjunction missing before „impacts“, i.e., “and”, „which“ or „that“? Or divide it into two sentences for easier understanding.

L.100 and L. 203-204: I suggest replacing the expression “with promise” by “showing promise” or “promising”.

L.104: The item “marriage” appears standing out in this list of terms, because it is very specific while the others are rather general. I would prefer to read “ceremonies” or similar.

L.113: I suggest inserting a comma before “given”.

L.137-138: I might not understand it correctly, but what is the difference here between music and music domains?

L. 141: Please insert a comma before “and”.

L.215-216: I wonder why some nouns in this phrase are upper-case and if it makes sense in that context and sentence structure.

L.263-264: I think it is sufficient to use “detail” only once in the whole sentence for easier reading.

L.268: I wonder how “higher-level” music skills are defined?

L.376: Correctly, the German nouns Hund and Drei should be capitalized.

L.376: “we use” instead of “we us”.

L.512: I wonder if a 2-week window after training completion is too large and if effects already go missing after such time span. I do agree that the tests are time consuming and difficult to organize. Is there any rationale that the effects can be expected to last or for how long, especially after not continuing the training and not having done such a training ever before? How long do you estimate it will take to complete a single macro test?

L.552ff: Are the research assistants in choir and audio groups the same or different people? Could this be relevant regarding bias due to uneven personal engagement in one of the interventions?

L. 678-680: Please indicate what these persons are acknowledged for, i.e. what their contributions were.

7. PLOS authors have the option to publish the peer review history of their article (what does this mean?). If published, this will include your full peer review and any attached files.

Reviewer #1: No

Reviewer #2: No

Reviewer #3: **Yes: **Marie Köberlein

---

## [Author Response · Author response to Decision Letter 0]

11 Oct 2024

Reviewer #1: 

Important note: This review pertains only to ‘statistical aspects’ of the study and so ‘clinical aspects’ [like medical importance, relevance of the study, ‘clinical significance and implication(s)’ of the whole study, etc.] are to be evaluated [should be assessed] separately/independently. Further please note that any ‘statistical review’ is generally done under the assumption that study specific methodological [as well as execution] issues are perfectly taken care of by the investigator(s). This review is not an exception to that and so does not cover clinical aspects {however, seldom comments are made only if those issues are intimately / scientifically related & intermingle with ‘statistical aspects’ of the study}. Agreed that ‘statistical methods’ are used as just tools here, however, they are vital part of methodology [and so should be given due importance]. I look at the manuscript in/with statistical view point, other reviewer(s) look(s) at it with different angle so that in totality the review is very comprehensive. However, there should be efforts from authors side to improve (may be by taking clues from reviewer’s comments). Therefore, please do not limit the revision only (with respect) to comments made here.

COMMENTS: Because the study & manuscript is excellent in all aspects, there is hardly anything to point-out. However, I have different opinion regarding a few (only two) issues which are given below:

When lines 599-602 it is stated that “Given the sample size and random assignment to group, we assume that covariates such as age, gender, education, musical training/background, credibility, and expectation will be evenly distributed across the participants. Thus, we expect equivalence between participants and their randomly assigned intervention” then why further said “To test this assumption, we will use a well-known technique to test for equivalence, the Bayesian t-test”? Test proposed is appropriate and reference quoted [Stefan AM, Gronau QF, Schönbrodt FD, Wagenmakers EJ. A tutorial on Bayes Factor Design Analysis using an informed prior. Behav Res Methods. 2019;51: 1042–1058] is nice. However, it is often very rightly advised that when random allocation/assignment is used/done any statistical comparison of baseline characteristics is not necessary. In this context, I request authors to note that even if P-value(s) turn(s) out to be significant (while comparing baseline characteristics despite random allocation), it is, by definition, a false positive as you then are supposed to be testing ‘randomization’ then, which in any single trial may not balance all baseline characteristics {though I am sure that these learned authors already know these things}.

Please note that any regression techniques are not basically/originally developed for any sort of [between or within group(s)] comparison(s). Using mixed-model regression is very appropriate in given situation, but note that ‘Head-to-head’ comparison is expected, as this is an indirect/secondary/by-product testing, in my opinion.

As pointed out in ‘important note’ above “This review pertains only to ‘statistical aspects’ of the study and so ‘clinical aspects’ should be assessed separately/independently. In my opinion, to make this article acceptable (which is quite possible and easy), a small amount of re-vision (re-drafting) may be needed. ‘Minor revision’ is recommended.

Response: Thank you for your comprehensive comments on the statistical aspects of our study. It is important to note that the Bayesian t-test does not produce a p-value. The output of the Bayesian t-test is the Bayes Factor, which is the ratio of the marginal likelihood of the two models (or samples in this circumstance) being compared. In this case, the Bayes factor tells us the likelihood that the two samples are equivalent. We acknowledge that due to random sampling we can assume that we are likely to get a normal distribution of specific characteristics (i.e., musical training). However, our intention behind the equivalence testing is to provide statistical proof of this assumption, and if required, provide an additional justification for the use of a covariate in our models.

Reviewer #2: 

The authors present a Registered Report Protocol on a randomized and controlled study on the effect of group singing on speech perception in noisy surroundings and psychosocial health in a cohort comprising 210 participants (including controls).

The research direction, the paradigms used, and the analysis presented are excellent. The authors gave a comprehensive overview in the introduction section, considering plenty of topics and references recently published and motivating their project. Also, the method section is comprehensively written and seems reproducible.

Furthermore, this study generally meets the criteria to be considered for publication in Plos One. Nevertheless, some points should be addressed before being considered for publication.

General

• Please avoid some acronyms/abbreviations, such as HA, MSB, TMCM, etc., that were only used in a particular paragraph or section. Using these abbreviations makes it hard to read the text and is not necessarily needed.

Thank you – we have removed the abbreviation of HA, MSB, TMCM. We have retained OPERA and PRISM, which we argue are the better-known terms when compered to their full forms.

• Fig. 2 shows no additional relevant information and should be eliminated.

We have removed this figure.

• I miss a paragraph on „Expected results“. The statements in l.258-265 are more general. Can you give numbers on expected outcomes?

Unfortunately, we are limited in being able to provide metrics on expected outcomes. To the best of our knowledge, there is not study that has investigated this population with this intervention. The closest study would likely be Dubinsky et al., (2019) – but this was with hearing aid users. We believe our approach of being broad and general is appropriate and the best we can currently do. Hopefully this study will provide additional data for future studies and their predictions.

Introduction

• l.252: Please replace MSP with MSB (but see my comment on abbreviations before)

Thank you for noticing this error. It has been replaced with the music and social bonding hypothesis in full.

Methods and Methods

• l. 376: hund -> Hund

• l.376: us -> used

Thank you for noticing these two spelling errors. They have been corrected.

• l.339 & l.521: Hearing loss-related inclusion criteria will be evaluated at four frequencies of 500, 1000, 2000, and 4000 Hz, but audiological assessment at 1000, 2000, 4000, and 8000 Hz. Why this discrepancy? This discrepancy is especially relevant for older patients, where hearing loss affects higher frequencies.

Again – thank you for noticing this discrepancy! This has been corrected to the standard 500 Hz, 1000 Hz, 2000 Hz, and 4000 Hz for both paragraphs.

Data Analysis

• l.609/610: Please carefully check the syntax of Eq. (1). I suppose „min“ should be subscripted.

Thank you – this has been updated and correctly subscripted.

• l.671-673: Please avoid R-syntax. It would be more helpful to explain what will be dependent variables and which explanatory variables (and their interaction) will be considered.

The syntax has been removed and the full form reads as:

To do this, a hierarchical method will be used to compare a model which predicts speech perception difference score from BAT, FDL, and SMRT difference scores to a model which also contains an interaction term between speech perception variables and the group variable.

Reviewer #3: 

This is a thorough and well-planned study regarding a promising subject.

ARHL affects large parts of the population and brings with it further serious life limitations on social, emotional and cognitive levels. Group singing as culturally anchored activity could be a simple and inexpensive method to counteract the negative consequences of ARHL, and there is already some evidence of effectiveness. However, previous studies have flaws that do not allow any watertight conclusions to be drawn. The presented study attempts to generate clarity through a large number of test subjects and control groups, the multi-site approach and extensive multi-level testing and statistics. The study meets the PLOS ONE publication criteria.

I have only a few remarks listed below.

General: Regarding the central point of the study, I would appreciate a little more detailed outline of the group singing sessions. It is totally legitimate that there can (must!) be flexibility and adjustments in the sessions’ progressions and I acknowledge that each protocol will be supplemented in the final report. However, since there are many different approaches for choir/group singing, it would be interesting to know a few conceptual details, e.g. whether there is going to be a warm-up of which duration and content or what repertoire is roughly in focus. What are the choir leaders’ qualifications/specifications?

Thank you for this comment. We have expanded some of the details which are presented in the full paragraph:

“Group singing will be facilitated by a choir master with at least one-year of experience leading a choir, a piano accompanist, and support from up to two research assistants who will sing with the participants. The broad goal of the group singing is to provide synchronous vocal pitch training in a supportive and casual environment. The beginning of each session will involve approximately 15-minutes of warm-up exercises, and the repertoire will be decided through discussion between the choir master and participants. Given the choir master will have some flexibility and need to adapt the sessions depending on group ability, a full outline of each lesson will be provided as supplementary materials in the final report.”

We have also expanded upon the choir master requirements to specify they must have at least one-year of experience leading a choir. However, we are hesitant to apply any additional qualification to this. While many choir masters have a educational background in a music-related undergraduate degree or higher, we do not believe this is a necessary requirement.

Title: The title seems very long – is there an possibility to shorten it?

Yes – we agree that it is quite long. However, the recommendations from the SPIRIT checklist are to provide a descriptive title – we have tried to be as brief as possible, but for such a large and comprehensive study, I hope you can understand the necessity of the length to sufficiently adhere to the SPIRIT guidelines.

L. 31: The word „appropriate“ is somewhat undefined: Could you insert something that makes clear what it means in this context?

Agree – we have redefined this as “an appropriately powered sample size”.

L. 52: „Older adults“ seems undefined. I would appreciate if you gave their actual age range, where possible, i.e., 60+

Agree – we have specified this as “adults aged 60 years and older.”

L. 61: Is there a conjunction missing before „impacts“, i.e., “and”, „which“ or „that“? Or divide it into two sentences for easier understanding.

Thank you for noticing this omission – we have added “that” to make the sentence grammatically correct.

L.100 and L. 203-204: I suggest replacing the expression “with promise” by “showing promise” or “promising”.

These changes have been made, with each sentence reading as:

“One potential non-clinical intervention showing promise is group singing.”

“Overall, these findings indicate that group singing is an enriching activity that shows promise.”

L.104: The item “marriage” appears standing out in this list of terms, because it is very specific while the others are rather general. I would prefer to read “ceremonies” or similar.

Yes – happy to make the change to ceremonies.

L.113: I suggest inserting a comma before “given”.

Thank you for the suggestion. After reading and adjusting, we decided to change it to “as” – without the need for a comma: “A potential benefit of group singing is the enhancement of speech perception, which is particularly relevant as ARHL is primarily associated with difficulties perceiving SIN.”

L.137-138: I might not understand it correctly, but what is the difference here between music and music domains?

No distinction in this case, simply an error – this has been corrected to speech and music domains.

L. 141: Please insert a comma before “and”.

Done.

L.215-216: I wonder why some nouns in this phrase are upper-case and if it makes sense in that context and sentence structure.

Thank you for pointing this out – we have adjusted these, so they are all consistent with a lower-case form.

L.263-264: I think it is sufficient to use “detail” only once in the whole sentence for easier reading.

Agree – we have removed the second instance.

L.268: I wonder how “higher-level” music skills are defined?

We have clarified that the Music-in-Noise Task (MINT) is our conceptualization of a higher-level music task in line 423. “The Music-in-Noise Task (MINT) will be used to measure inter-individual differences in hearing-in-noise skills [110]. MINT is utilized as a measure of higher-level musical skills.”

L.376: Correctly, the German nouns Hund and Drei should be capitalized.

Thank you – this has been corrected.

L.376: “we use” instead of “we us”.

Done.

L.512: I wonder if a 2-week window after training completion is too large and if effects already go missing after such time span. I do agree that the tests are time consuming and difficult to organize. Is there any rationale that the effects can be expected to last or for how long, especially after not continuing the training and not having done such a training ever before? How long do you estimate it will take to complete a single macro test?

We are operating under the assumption that the effect of the intervention will have sufficient impact that will last longer than a week after the cessation of training. The 2-week window is a pragmatic constraint that factors in each sites’ capacity to effectively manage the testing of up to 30 participants (if they run both the group-singing and audiobook interventions concurrently). We estimate that each macro test session will take approximately 1.5 hours. The 2-week allocation provide the research team a more achievable target as we are operating under robust, registered protocol approach (i.e., we can state quite confidently all participants will be testes within this period – we are much less confident if we use a 1-week constraint). Additionally, it also factors in participant availability by providing them more testing appointment times. Having completed a number of longitudinal intervention studies – these pragmatic challenges are important to consider.

L.552ff: Are the research assistants in choir and audio groups the same or different people? Could this be relevant regarding bias due to uneven personal engagement in one of the interventions?

There is no requirement, and some sites may choose to utilize the same research assistants, and other different ones. Much of this may come down to each sites capacity. As the RAs are providing a supporting role compared to the Choir Master or Audiobook Facilitator, and also in comparison to the actual intervention itself, we do not believe this will have a significant impact or bias. 

L. 678-680: Please indicate what these persons are acknowledged for, i.e. what their contributions were.

This has been added: “We thank the following individuals and organizations for providing support and advice for this project:”

---

## [Editor Report · Decision Letter 1]

12 Nov 2024

Speech-in-noise, psychosocial, and heart rate variability outcomes of group singing or audiobook club interventions for older adults with unaddressed hearing loss: a SingWell Project multisite, randomized controlled trial, registered report protocol

PONE-D-24-30955R1

Dear Dr. Lo,

We’re pleased to inform you that your manuscript has been judged scientifically suitable for publication and will be formally accepted for publication once it meets all outstanding technical requirements.

Kind regards,

Michael Döllinger, Ph.D.

Academic Editor

PLOS ONE

---

## [Editor Report · Acceptance letter]

21 Nov 2024

PONE-D-24-30955R1 

PLOS ONE

Dear Dr. Lo, 

I'm pleased to inform you that your manuscript has been deemed suitable for publication in PLOS ONE. Congratulations! Your manuscript is now being handed over to our production team.

Kind regards, 

on behalf of

Dr. Michael Döllinger 

Academic Editor

PLOS ONE